METHODS AND RESOURCES

# A cloud-based toolbox for the versatile environmental annotation of biodiversity data

**Richard Li**[1,2], **Ajay Ranipeta**[1,2], **John Wilshire**[1,2], **Jeremy Malczyk**[3], **Michelle Duong**[1,2], **Robert Guralnick**[4], **Adam Wilson**[5], **Walter Jetz**[1,2]\*

1 Department of Ecology and Evolutionary Biology, Yale University, New Haven, Connecticut, United States of America, 2 Center for Biodiversity and Global Change, Yale University, New Haven, Connecticut, United States of America, 3 Descartes Labs, Santa Fe, New Mexico, United States of America, 4 Florida Museum of Natural History, University of Florida, Gainesville, Florida, United States of America, 5 Department of Geography, University at Buffalo, Buffalo, New York, United States of America

\* walter.jetz@yale.edu

**Data Availability Statement:** Data and code from this manuscript's case studies are located on GitHub, archived through Zenodo: https://doi.org/10.5281/zenodo.5208219 ".

## Abstract

A vast range of research applications in biodiversity sciences requires integrating primary species, genetic, or ecosystem data with other environmental data. This integration requires a consideration of the spatial and temporal scale appropriate for the data and processes in question. But a versatile and scale flexible environmental annotation of biodiversity data remains constrained by technical hurdles. Existing tools have streamlined the intersection of occurrence records with gridded environmental data but have remained limited in their ability to address a range of spatial and temporal grains, especially for large datasets. We present the Spatiotemporal Observation Annotation Tool (STOAT), a cloud-based toolbox for flexible biodiversity–environment annotations. STOAT is optimized for large biodiversity datasets and allows user-specified spatial and temporal resolution and buffering in support of environmental characterizations that account for the uncertainty and scale of data and of relevant processes. The tool offers these services for a growing set of near global, remotely sensed, or modeled environmental data, including Landsat, MODIS, EarthEnv, and CHELSA. STOAT includes a user-friendly, web-based dashboard that provides tools for annotation task management and result visualization, linked to Map of Life, and a dedicated R package (*rstoat*) for programmatic access. We demonstrate STOAT functionality with several examples that illustrate phenological variation and spatial and temporal scale dependence of environmental characteristics of birds at a continental scale. We expect STOAT to facilitate broader exploration and assessment of the scale dependence of observations and processes in ecology.

## Introduction

Spatiotemporal biodiversity data are accumulating rapidly, offering an ever larger and more diverse foundation for research. Citizen science observations are driving much of this increase [1,2], complemented in recent years by GPS tracking data and camera trapping data [3,4]. Simultaneously, there have been substantial increases in the quantity, spectral, and

**Funding:** We acknowledge funding from the NSF grant DEB-1441737 to WJ and RG and support from NASA grants 80NSSC17K0282 and 80NSSC18K0435 to WJ, RG, and AW. The funders had no role in study design, data collection and analysis, decision to publish, or preparation of the manuscript.

**Competing interests:** The authors have declared that no competing interests exist.

**Abbreviations:** AppEEARS, Application for Extracting and Exploring Analysis Ready Data; Env-DATA, Environmental Data Automated Track Annotation System; EVI, Enhanced Vegetation Index; LST, Land Surface Temperature; NASA, National Aeronautics and Space Administration; NDVI, Normalized Difference Vegetation Index; STOAT, Spatiotemporal Observation Annotation Tool.

spatiotemporal resolution and availability of ecologically relevant environmental data, thanks to increasing Earth-orbiting satellite-based sensors and powerful models of global environmental conditions [5–7].

The fusion of these 2 data types, that is, the characterization of environmental conditions at, or surrounding, spatiotemporal points or polygons representing biodiversity data, is commonly called "environmental annotation" of biodiversity records (e.g., [8]). Annotations may range from simple space-time intersections between biodiversity and environmental data, to more complex zonal calculations incorporating values from neighboring space-time locations, analogous to feature extraction in purely spatial applications. Environmental annotation of biodiversity records over larger scales has already supported a broad range of applications and insights. These include studies of species distribution [9,10], animal movement [8,11,12], phenology [13,14], habitat use [15,16], landscape ecology [17], disease ecology [18], climate change [19], and others. The continued growth in the volume and variety of biodiversity and environmental data is poised to further increase the scope of research applications.

## Matching the respective scales of observations, uncertainty, and process

Despite the large volume and breadth of research relying in some form on the combination of biodiversity and environmental data, best approaches for proper integration have yet to be well specified. Notably, both spatiotemporal biodiversity and environmental data vary widely in their associated spatial and temporal grains and uncertainties [20]. Biodiversity records may refer to split-second observations of single organisms or larger areas sampled over months. They may be measured with extremely high accuracy, as for GPS-based tracking or camera trap records [3,21], or be less precise in their time stamps or locations as is often the case for citizen science or older museum specimen data. Environmental data layers similarly range from precisely captured 30-m pixels overflights to monthly 1-km characterizations. Beyond these discrepancies in the spatiotemporal attributes of data, there are well-recognized temporal and spatial scale dependencies in ecological processes [22,23]. Connecting environmental and biodiversity observation data in a way that does not address the respective spatial or temporal scales (i.e., grains) of evidence or process can severely compromise ecological inference and prediction. Currently, many studies relying on environment—biodiversity data annotation do not actively consider these scale issues [12], or do so only on an ad hoc basis [11], thus incurring observation and process mismatches and missed opportunities for more effective and informed integration.

We suggest that a central reason for the relatively limited consideration of these data and process scale differences, despite rapidly growing data, is the lack of performant, readily usable tools. This is the motivation for the development of the versatile biodiversity–environment annotation toolbox that we present here: Spatiotemporal Observation Annotation Tool (STOAT). To further support STOAT's relevance and specific uses, we explore 3 major concepts in detail (Fig 1): (i) spatiotemporal observational grain (data resolution); (ii) spatiotemporal uncertainty; and (iii) spatiotemporal process scale.

**Spatiotemporal observational grain.** The grain of observations and environmental data can vary widely depending on their sources and subsequent processing. A species observation could range from a single point in time and space where an individual was recorded by a GPS collar, to a polygon with an area of thousands of square kilometers representing a national park the species was known to inhabit over the past few decades. Likewise, an environmental characterization of a location could range from an estimate of vegetation status on a specific day for an area of $25m^2$ to a multidecadal mean averaged over hundreds of $km^2$. The observations are essentially the same (species location or vegetation status), but the grain and associated interpretation of those observations are, or should be, very different.

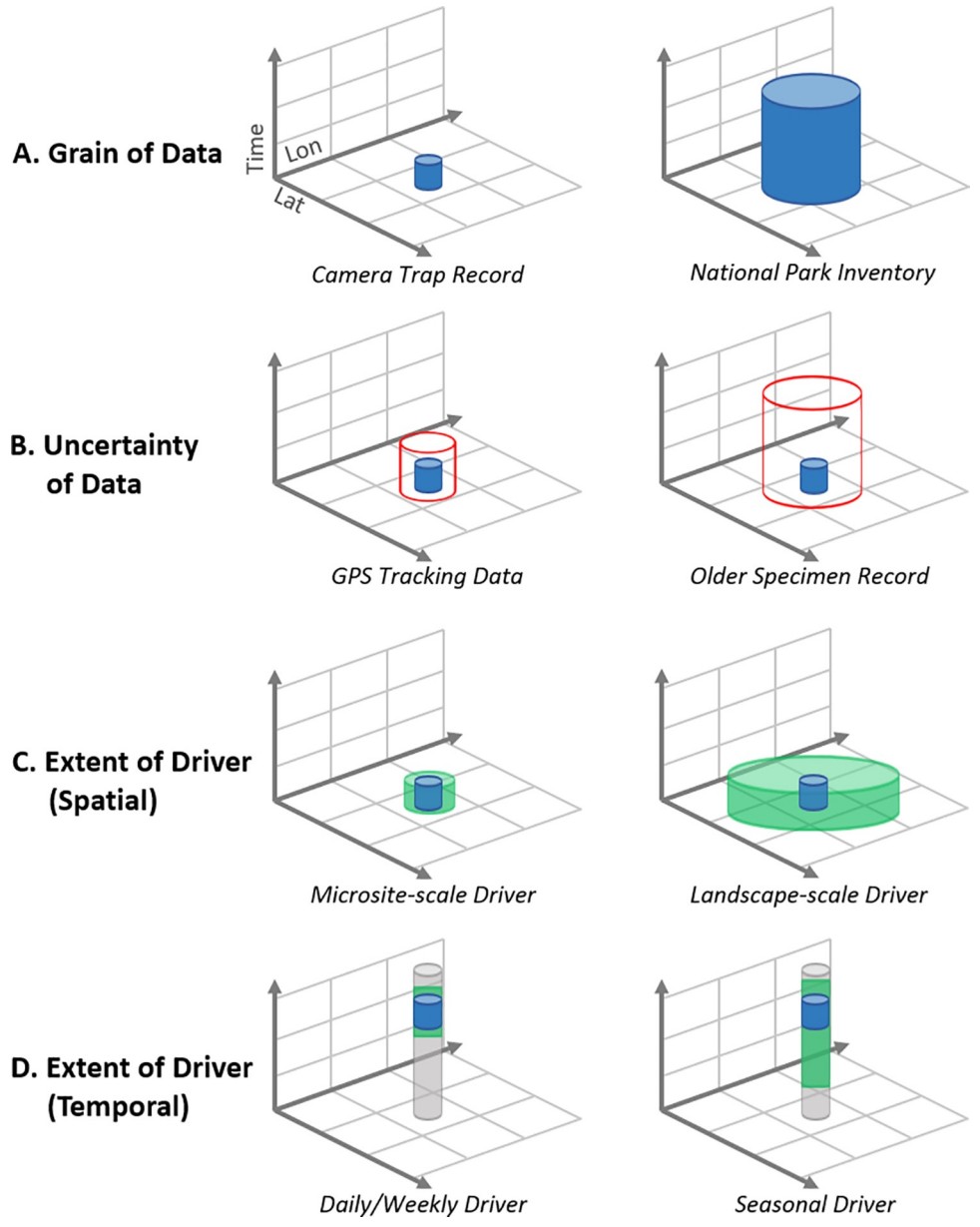

**Fig 1.** Variation in spatiotemporal characteristics of **(A, B)** biodiversity data and **(C, D)** ecological processes underlying biodiversity data. The blue cylinders reflect the spatiotemporal grain of the data itself, with the red and green cylinders reflecting data uncertainty and driver extent, respectively.

The linking of environmental with biodiversity data adds a further set of challenges. The grains of environmental data are predetermined and may or may not align with those of biodiversity data [6]. Likewise, the grains of biodiversity data may differ from each other in analyses that aggregate data from multiple sources. The intersection of spatially fine-grain biodiversity data with coarser gridded datasets is also vulnerable to the modifiable areal unit problem [24]. When considering grain heterogeneity, entire classes of biodiversity data may be unfit for simple intersections with categories of environmental products: temporally fine-grain camera trap observations with temporally coarse climatological products and spatially coarse biological survey data with fine-grain remote sensing products.

The 2-way spatiotemporal grain variability between biodiversity observations and environmental data increases the likelihood of mischaracterization following a simple intersection (i.e., identifying the grid cell of environmental data that is closest to a biodiversity observation). The impact of mismatches in data grain size can be lessened using techniques such as data coarsening or statistically using multiscale models [6,25], but all of these methods presume existing knowledge of biodiversity and environmental data grain as well as the grain of the relevant ecological processes, in addition to access to the appropriate data. Furthermore, coarsening either type of data risks removing fine-grain variability that is critically important for understanding the underlying ecological processes. Data coarsening loses utility if the spatiotemporal scale of an ecological process is finer than the finest grain biodiversity or environmental data available. Demand for data of higher resolution for the study of fine-scale ecological problems (e.g., animal behavior [12] or mechanistic niche models [26]) demonstrates the need for advancement in our biodiversity and Earth observation technologies. Addressing grain mismatch is made more difficult by the general lack of attention to scale dependency [12,27]; spatial and temporal scale metadata of biodiversity data (especially in large aggregated databases) are often unavailable, lost, or discarded [28,29].

**Spatiotemporal uncertainty.** Biodiversity observations also vary in their spatiotemporal uncertainty owing to the variety of data collection methods. These can vary from a global positioning system location with spatiotemporal accuracies of seconds and meters to text-based descriptions of location common in museum collections, which may have uncertainties measured in kilometers and days or even years. The interpretation of uncertainty is distinct from that of grain. Consider 2 hypothetical biodiversity observations: the first with a coarse spatial grain but low uncertainty (e.g., a polygon representing the home range of a bird estimated from many GPS observations) and the second with fine grain but high uncertainty (e.g., a record of a museum specimen with imprecise geocoding). For the first GPS-derived range, taking the mean and variance of the environmental conditions across the range may be an appropriate way to characterize the individual's environmental niche. The environmental characterization of the museum observation could also utilize the mean and variance of the area of spatial uncertainty, but would be more appropriately interpreted as a random variable drawn from the possible values across the area of uncertainty. The organism recorded in the second case is not assumed to inhabit all locations within its area of uncertainty, and thus the same mean and variance can be interpreted differently.

**Spatiotemporal process scale.** Ecological processes are highly scale dependent, a fact that has been documented in analyses spanning a broad range of disciplines [23]. Data process mismatch occurs when the spatial or temporal scales (i.e., grain or extent) of attempted observations of an ecological process differ from the scales at which the process produces its signal. Depending on the severity of mismatch, the observations can produce a signal that is weaker than the true signal, masked by noise, or even opposite in directionality to the true signal [27]. Field observations are often used to generate metrics of biodiversity using data captured at different spatial scales, resulting in seemingly contradictory results across studies [30,31]. Considering the large scope of many studies using the present data fusion techniques, any mischaracterizations can lead to misguided conservation and management decisions that are based on inaccurate results. Researchers must account for the spatial and temporal scales of ecological processes and should make an effort to use data with appropriate scales during their data analyses. Biodiversity and environmental data suitability will vary from case to case: Long-term mean climatic conditions may be suitable for estimating the climatic niche of a long-lived plant species, but entirely unsuitable for understanding the movement dynamics of a long-distance bird migrant. Knowledge of one's particular research system is key: Insight

into the spatial or temporal scale of a studied process can help inform data selection, annotation, and assist in analysis and the interpretation of results.

## Enabling the versatile environmental annotation of biodiversity records

The pitfalls of the environmental annotation of data with varying grains, uncertainties, and process scales highlight the importance of scale explicitness, i.e., the careful consideration and documentation of scales of both biodiversity and environmental data and associated sampling or ecological processes. Ideally, such annotations would offer flexibility to assess sensitivity to necessary assumptions or uncertainties around these relevant scales. Operationally, zonal calculations in space and time around a point of interest, i.e., spatiotemporal "buffers," can help to change the effective grain of biodiversity data. Buffers can be used to match the grain of biodiversity data to that of an environmental layer or the scale of an ecological process. They can help account for spatiotemporal uncertainty in species records by averaging over potential occurrence locations. Spatiotemporal buffering is thus a versatile procedure with the potential to improve the environmental characterization of species for analyses that require species–environment matching. Easily scale-adjustable annotations would additionally support the exploration and quantification of scaling relationships themselves, both for studies primarily investigating spatiotemporal scaling effects and for studies focused on potentially scale-dependent ecological phenomena.

We introduce STOAT to fill the need for a performant, scale-explicit, and scale-adjustable environmental annotation service. STOAT provides buffering capability in both space and time, supporting explorations and decisions around the grain and uncertainty dependence of results and scale-variant processes. STOAT is optimized for geographically large or spatiotemporally heterogeneous datasets, such as global species occurrence records, and serves annotations against a varied and growing set of remote sensing–supported and other environmental layers at a range of spatiotemporal scales. We use several cases studies to demonstrate how STOAT can provide the specificity and versatility needed for more informed environmental annotation of biodiversity data.

With a combined focus on performance and scale versatility, STOAT complements existing environmental annotation tools. These include the R package *raster* [32], which enables annotation on local computers, but requires download and storage of environmental layers. For movement data, the Environmental Data Automated Track Annotation System (Env-DATA) in Movebank supports single point in time-specific environmental annotations with a particular view to behavioral analyses [8]. National Aeronautics and Space Administration (NASA's) Application for Extracting and Exploring Analysis Ready Data (AppEEARS) [33] provides annotation functionality for individual occurrence coordinates or geographic regions, and, more recently, the Google Earth Engine R package (*rgee*) [34] offers access Google Earth Engine's environmental layer catalog [35]. In all of these tools, further user scripting to support spatiotemporally versatile annotations is required, and they thus remain nonoptimized for that purpose.

STOAT can be accessed through a web client (https://mol.org/stoat), which offers a range of visual and map-based reports, and dashboards for the management of annotation tasks and data. An associated R package (*rstoat*) provides core functionality in the R environment. STOAT is integrated with Map of Life (https://mol.org), a platform supporting the analysis and use of species distribution information, and its biodiversity mapping and analysis tools. The goal of STOAT is to support easier, more flexible, and scale-conscious integration of environmental and biodiversity data.

## Methods and features

### STOAT functionality and usage

An overview of usage steps for the web application (https://mol.org/stoat) and the R package (*rstoat*) is given through the website, associated help files, and an R vignette. In the current version of STOAT (v1.0), user datasets for annotation are uploaded into a private cloud datastore (large datasets) or can be loaded into memory in R (small datasets). The current version accepts occurrence records structured in a comma-separated values (.csv) file, with other data types such as polygons planned for the future. STOAT v1.0 limits environmental annotation jobs in size and layer count as a pragmatic way to avoid time and compute costly operations. Current limits can be found on the STOAT website. Computationally highly intensive annotations are possible (the architecture is fully scalable), but in v1.0 require coordination of cloud computing resources with the STOAT team. STOAT is designed to efficiently handle global datasets and is thus well situated to link with GBIF and other large spatiotemporal biodiversity datastores.

After biodiversity data are uploaded, users are able to select annotation options regarding the biodiversity dataset, environmental products, and spatial and temporal buffers. These selections can be made from either the web interface (Fig 2) or functions available in the R package. Fig 3 shows the overall architecture of STOAT, explored in detail below. The various backend steps of the annotation process (Fig 4) are executed using a cloud-based computing

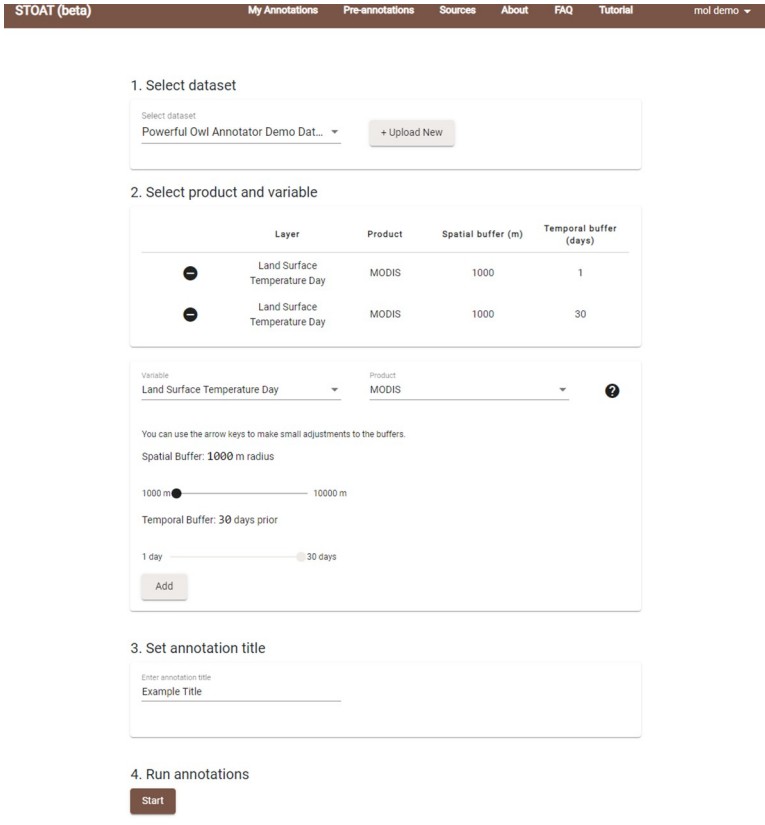

**Fig 2. STOAT environmental annotation dashboard.** Here, a test dataset is set up for annotation against the MODIS Day LST layers. A constant 1,000-m buffer is used, reflecting the native resolution of the environmental data, whereas temporal buffers are varied. LST, Land Surface Temperature; STOAT, Spatiotemporal Observation Annotation Tool.

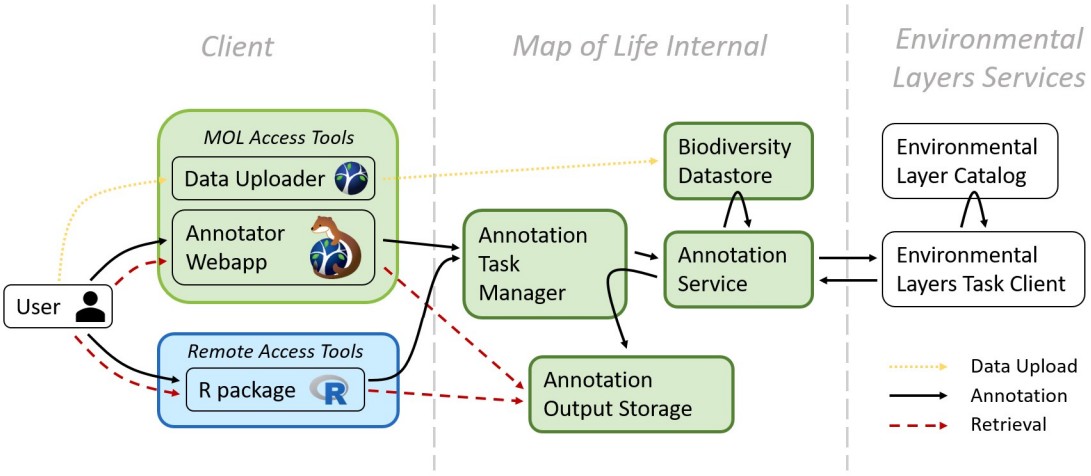

**Fig 3. STOAT architecture.** STOAT, Spatiotemporal Observation Annotation Tool.

cluster, obviating the need for users to download large environmental datasets, especially important when working with high spatial or temporal sampling frequencies. Upon the completion of the annotation job, users are notified via email and can keep results on the cloud or retrieve results through the STOAT web application or R package. STOAT returns a ZIP file containing (i) comma-separated values (.csv) files with the annotated values and associated statistics; and (ii) text (.txt) files containing the parameters of the annotation, as well as metadata and layer citations. For small datasets, an "on-the-fly" annotation that provides results through the R console is available and allows users to run pilots for larger annotation jobs. *rstoat* is available through CRAN: https://CRAN.R-project.org/web/packages/rstoat/index.html. For the code base, see https://github.com/MapofLife/rstoat.

## Environmental datasets

STOAT offers key global gridded environmental datasets at their native spatial and temporal resolutions. These environmental data span a variety of environmental characterizations, including climate, topography, vegetation, land cover, and human impacts. We group these datasets according to their temporal frequencies of sampling. These range from static products (single layer typically representing a long time period) to annual products, monthly products, and submonthly (dynamic) products.

Dynamic products in v1.0 include MODIS day/night Land Surface Temperature (LST) at 1-km spatial resolution and daily temporal resolution. MODIS vegetation indices (i.e., Enhanced Vegetation Index [EVI] and Normalized Difference Vegetation Index [NDVI]) are provided at 250-m spatial and daily temporal resolution. STOAT also provides vegetation indices for both Landsat 7 and Landsat 8, each at a 30-m spatial and 16-day temporal resolution. Finally, STOAT provides access to the EarthEnv-CHELSA high-resolution precipitation dataset [36] at 1-km spatial and daily temporal resolution.

Other layers provided on release include the European Space Agency's global land cover maps [37], global human modification by the Nature Conservancy [38], global climatologies from CHELSA [39,40], and selected layers from the EarthEnv project (https://earthenv.org), including topography [41], habitat heterogeneity [42], and cloud cover [43]. As additional layers are developed, they will become available through STOAT; for the v1.0 list, see Table 1, and consult https://mol.org/stoat/sources for up-to-date information.

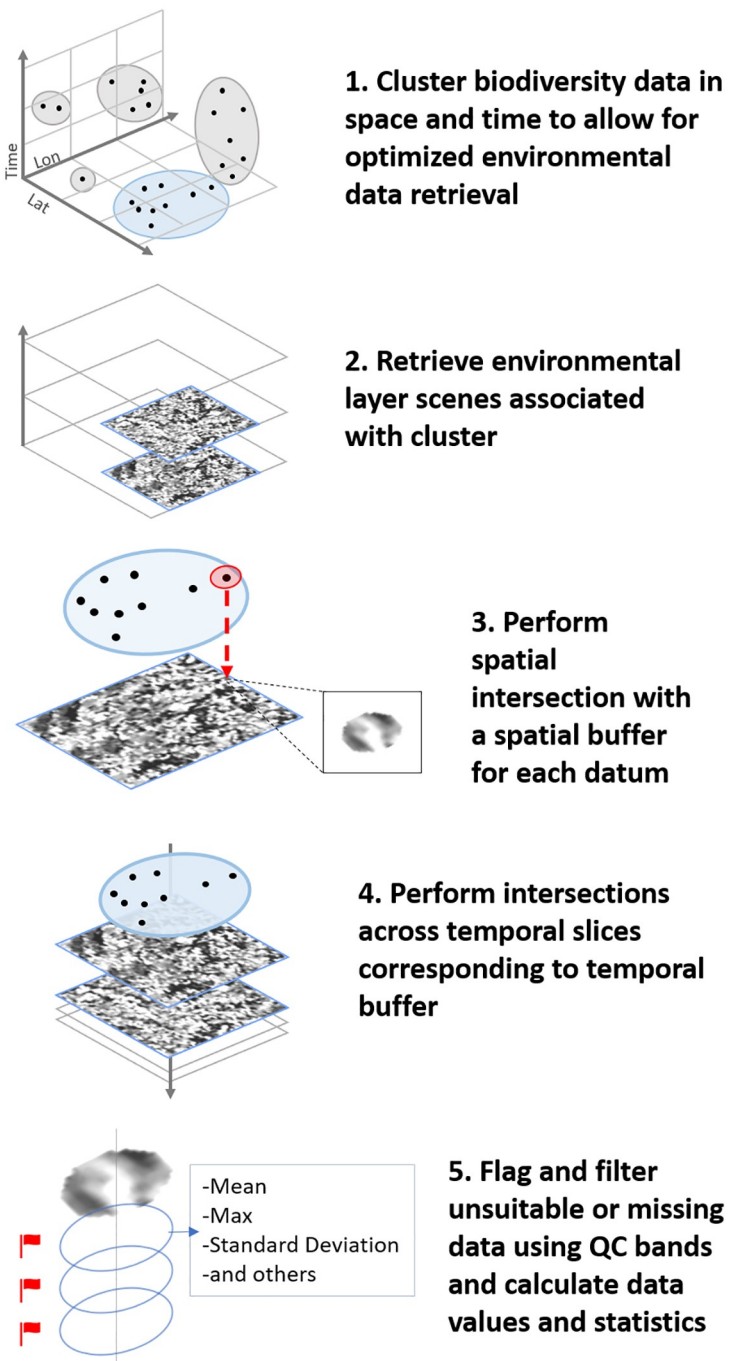

1. Cluster biodiversity data in space and time to allow for optimized environmental data retrieval

2. Retrieve environmental layer scenes associated with cluster

3. Perform spatial intersection with a spatial buffer for each datum

4. Perform intersections across temporal slices corresponding to temporal buffer

5. Flag and filter unsuitable or missing data using QC bands and calculate data values and statistics

- Mean
- Max
- Standard Deviation
- and others

**Fig 4. STOAT environmental annotation workflow. Clustering procedure (Step 1) is carried out only for occurrence datasets of sufficient size.** QC, Quality Control; STOAT, Spatiotemporal Observation Annotation Tool.

## Custom spatial and temporal grains (buffers)

STOAT allows users to manipulate effective biodiversity data grain size using its user-customizable spatial and temporal buffers. STOAT draws spatial buffers as a radius around the

**Table 1. Environmental products provided for annotation in STOAT v1.0 (consult https://mol.org/stoat/sources for latest).**

| Category | Product name | Product source | Available layers | Spatial coverage | Temporal availability | Spatial grain | Temporal grain | Citation |
|---|---|---|---|---|---|---|---|---|
| Dynamic | MODIS (M*D09) | NASA (https://lpdaac.usgs.gov) | EVI, NDVI | Global | 2000/02/24 to present | 250 m | Daily | http://doi.org/10.5067/MODIS/MOD09A1.006 http://doi.org/10.5067/MODIS/MYD09A1.006 |
| Dynamic | MODIS (M*D11A1/A2) | NASA (https://lpdaac.usgs.gov) | LST Day, LST Night | Global | 2010/01/01 to present | 1,000 m | Daily | http://doi.org/10.5067/MODIS/MOD11A1.006 |
| Dynamic | Landsat 7 | NASA/USGS | EVI | Global | 1999/05/01 to 2017/04/17 | 30 m | 16 days | https://www.usgs.gov/core-science-systems/nli/landsat |
| Dynamic | Landsat 7 Real Time Collection 1 | NASA/USGS | EVI | Global | 2017/05/02 to present | 30 m | 16 days | https://www.usgs.gov/core-science-systems/nli/landsat |
| Dynamic | Landsat 8 | NASA/USGS | EVI | Global | 2013/04/11 to 2017/04/23 | 30 m | 16 days | https://www.usgs.gov/core-science-systems/nli/landsat |
| Dynamic | Landsat 8 Real Time Collection 1 | NASA/USGS | EVI | Global | 2017/05/06 to present | 30 m | 16 days | https://www.usgs.gov/core-science-systems/nli/landsat |
| Dynamic | CHELSA/EarthEnv Daily Precipitation | CHELSA | Precipitation | Global | 1979/01/01 to 2018/11/30 | 1,000 m | Daily | Karger and colleagues (in review) |
| Annual | ESA CCI | ESA (https://www.esa-landcover-cci.org/) | Land cover | Global | 1992 to 2018 | 300 m | Annual | ESA, 2017 |
| Static | SRTM | NASA (https://lpdaac.usgs.gov) | Altitude, aspect, and slope | 60˚N—56˚S 180˚E—180˚W | | 30 m | Static | http://doi.org/10.5067/MEaSUREs/SRTM/SRTMGL1.003 |
| Static | CHELSA Static Layers | CHELSA (http://chelsa-climate.org/downloads/) | Mean annual temperature, precipitation, and others | Global | | 1,000 m | Static | Karger and colleagues (2017) and Karger and colleagues (2018) |
| Static | EarthEnv: Topography | EarthEnv (https://www.earthenv.org/topography) | Topographic Position Index and Terrain Ruggedness Index | Global | | 1,000 m | Static | Amatulli and colleagues (2018) |
| Static | EarthEnv: Habitat Heterogeneity | EarthEnv (https://www.earthenv.org/texture) | Coefficient of variation and Homogeneity | Global | | 1,000 m | Static | Tuanmu and Jetz (2015) |
| Static | EarthEnv: Cloud Cover | EarthEnv (https://www.earthenv.org/cloud) | Intra-annual variability | Global | | 1,000 m | Static | Wilson and Jetz (2016) |
| Static | TNC Global Human Modification | TNC (https://figshare.com/articles/Global_Human_Modification/7283087) | Human modification | Global | | 1,000 m | Static | Kennedy and colleagues (2019) |
| Static | Derived Seasonal EVI | Derived layer generated from MODIS EVI | Winter EVI (November to February) and Summer EVI (June to August) | Global | | 1,000 m | Static | |

EVI, Enhanced Vegetation Index; LST, Land Surface Temperature; NDVI, Normalized Difference Vegetation Index.

coordinate(s) of interest, aggregating environmental layer pixel values within the buffered region to generate the output values including the mean, standard deviation, and the number of pixels present in the buffer. Only pixels within the buffered area are included (as determined by GDAL's rasterize function). Temporal buffers are implemented by aggregating values across the days before the date of interest after the spatial buffers have been applied at the daily

level (including the day of observation, so minimum is 1). These spatial and temporal buffers allow the user to adjust the effective grain size of the environmental data, characterize uncertainty, and match the grain of the data with the underlying ecological processes.

The effective use of both spatial and temporal buffers requires basic knowledge of the spatial and temporal grains of the environmental product requested. For example, a temporal buffer of 7 days on a MODIS (daily resolution) request would generate 7 days of data (the day of and 6 days before), but might not generate a single value when applied to a 16-day Landsat sampling interval. Suggested buffers for various products are provided as defaults to allow basic data accessibility to those less familiar with the underlying datasets.

## Architecture and workflows

STOAT leverages software and infrastructure developed in Map of Life. The STOAT architecture consists of 3 units: the STOAT client, server, and environmental layer services (Fig 3). Users launch tasks through the STOAT client (via the web app or R), including data uploads, annotation jobs, and result retrievals. Upon receipt of an annotation request, the STOAT server (annotation task manager) schedules the annotation job and allocates computing resources. The annotation service then initiates and connects to both the biodiversity datastore (where data for annotation are stored) and the environmental layer service. The annotation service sends tasks for initiation of the annotation workflow (Fig 4) to the layer service, for annotation against the layer catalog. Results are returned to the annotation service and stored for later retrieval by the user. At launch (v1.0), environmental layers are provided by Descartes Labs, a geospatial data aggregator and analytics platform provider (https://descarteslabs.com), and Google Earth Engine [35]. Annotation jobs are managed by the Map of Life platform, which runs the annotation server, and stores uploaded and annotated biodiversity data.

STOAT is optimized for the environmental annotation of large numbers of spatiotemporal biodiversity records. Nevertheless, the computational requirements for annotations with large spatial and temporal buffers can be substantial. Computation time in practice depends on the number of records, the size of the spatial and temporal buffers, the number and temporal density of layers for annotation, and the availability of computational resources. For an overview of computational benchmarks for STOAT v1.0, see S1 Table.

## Annotations at scale

STOAT uses occurrence clustering to support the efficient annotation of particularly large biodiversity datasets. By grouping records in space and time before annotation, STOAT can reduce the number of computationally expensive scene retrievals. Spatial clustering is implemented using geohashes, wherein occurrences are classified into standardized grid cells according to coordinate location [44]. Geohash length (i.e., precision) is dependent on the spatial grain of the environmental layer, with longer geohashes used for finer grain layers (e.g., Landsat). As time stamps are hierarchical in nature, temporal grouping is implemented by binning occurrences into their year, year-month, or year-month-week of observation, depending on the product temporal grain and the level of spatial clustering used. By spatially and temporally grouping records, STOAT can achieve significantly improved performance in areas with high record density (e.g., urban areas or citizen science hotspots). As an example, one could consider 500 records within a small area on a single date, for which a 7-day temporal buffer is requested. A simple iterative annotator would need to retrieve a scene 3,500 times. STOAT's clustering and buffer integration allow a user to complete the task with only 7 scene retrievals. Maximally time-efficient annotations enabled through clustering come at computational expense. Since clusters are annotated in parallel by computing nodes, the approach is only suitable when the number of

records annotated by a single node outweighs the cost of node use. However, clustering methods remain an option for very large or very spatially dense biodiversity datasets.

## Case studies

We present 2 case studies to highlight STOAT's research potential and to illustrate use cases for its grain-flexible environmental annotations. First, we compare the annual greenness and temperature niches of a migratory bird species (ruby-throated hummingbird, *Archilochus colubris*) with those of a nonmigratory species (tufted titmouse, *Baeolophus bicolor*). Using STOAT's spatiotemporal buffers, we demonstrate both spatial and temporal scale dependence of the 2 species' environmental niches. Second, we explore differences in the annotated greenness values of Anna's hummingbird (*Calypte anna*) occurrences across annotation methods. We compare the results of a record grain-specific environmental annotation facilitated by STOAT to a uniform environmental annotation representative of current popular methods.

**Spatial and temporal scale dependence of species niches.** To illustrate the STOAT workflow and the utility of scale-adjustable annotations, we chose 2 bird species native to North America: the migratory ruby-throated hummingbird (https://mol.org/en/species/map/ Archilochus_colubris) and the nonmigratory tufted titmouse (https://mol.org/species/map/ Baeolophus_bicolor). We downloaded global occurrences for the 2 species from eBird [45,46] and filtered the data using the *Auk* R package [47], for the years 2013 to 2017, for data collected under the "Stationary" protocol, and for coordinates associated with "Personal" locality types. Personal localities in eBird are occurrence locations manually provided by the observer and are of finer and less variable grain than eBird hotspots, which may range from the size of a city block to a national park. Data using the stationary protocol and personal localities should have the finest and least variable grain among all eBird observations (30-m according to protocol definition). We expect spatial uncertainty to also be on the order of tens of meters (determined by the accuracy with which observers pin personal localities on a map). With the necessary assumptions about data spatial grain and uncertainty, we estimate that our occurrence data are approximately accurate to the 100-m scale. Temporal grain was standardized by coarsening data to the day of observation. Temporal uncertainty in eBird is usually on the order of hours rather than days, and we do not expect it to impact results.

From the records remaining after applying these filters, we randomly sampled 20,000 occurrences, which we spatially thinned using the *spThin* R package [48] to reduce the impacts of spatial sampling bias, resulting in 5,185 hummingbird occurrences and 5,021 titmouse occurrences.

We annotated occurrences of both species with the MODIS LST Day (1,000 m, 1-day grain) and Landsat 8 EVI (30 m, 16-day grain) layers to illustrate variation in environmental association across species over time. We conducted annotations at fine and coarse spatiotemporal buffering levels to demonstrate the spatial (hummingbird) and temporal (titmouse) scale dependence of species environmental characterizations. The ruby-throated hummingbird was annotated against Landsat 8 EVI with a spatial buffer of 120 m (the estimated spatial grain of the data and a round multiple of the native Landsat grain of 30 m) and 990 m, and the tufted titmouse was annotated against MODIS LST with a temporal buffer of 1 and 30 days. In each case, the other buffer dimension (temporal for hummingbird, spatial for titmouse) was left at native grain. The hummingbird was further annotated with EarthEnv Habitat Heterogeneity (EVI-based Coefficient of Variation, 1,000 m grain, static). After job completion, annotated results were downloaded and analyzed in R [49]. Records with no value successfully annotated (due to missing or poor data quality) were omitted from the analyses. Since larger buffers are associated with greater likelihoods of retrieving valid data, only records successfully annotated at both buffering levels were retained for cross-buffer comparisons ($n$ = 3,162 and 1,766 for

hummingbird and titmouse, respectively). The relatively low rate of successful annotation is attributable primarily to missing data from cloud cover and highlights additional utility provided by temporal buffers.

To explore the spatial scale sensitivity of the hummingbird environmental niche, we calculated the cross-grain differences in EVI ($|\Delta EVI|$) as the absolute difference between the EVI value of a given observation annotated at a 120-m grain and that of the same observation at a 990-m grain (i.e., 120 versus 990m spatial buffer). Higher $|\Delta EVI|$ values would indicate greater, and lower values lesser dissimilarity between the 2 spatial buffer sizes (grains). We compared $|\Delta EVI|$ with habitat heterogeneity as we hypothesized that greater spatial landscape heterogeneity would result in greater impact of grain on calculated EVI. For the titmouse, we calculated $|\Delta LST|$ as the difference between the LST value of an observation annotated at 1-day grain and 30-day grain. We compared the $|\Delta LST|$ with the date of observation to observe seasonal differences in experienced temperature variability. All analyses used R version 3.6.0 [49].

**Record grain-specific versus grain-agnostic annotations.** To highlight the utility of record grain-specific annotations enabled by STOAT, we explored environmental characterizations of the Anna's hummingbird. The species was chosen for its frequency of observation in eBird as well as for its geographic distribution. The Anna's hummingbird (https://mol.org/en/species/map/Calypte_anna) inhabits western regions in North America, many of which are more arid and less EVI saturated than the eastern forests occupied by the ruby-throated hummingbird. As in the previous case study, we first downloaded species occurrences from eBird [45,46], and filtered using *Auk* [47], for the years 2013 to 2017 and for coordinates associated with personal localities. Instead of retaining species occurrences at a single spatial grain by filtering for "Stationary" eBird observation protocol, we sorted occurrences into different spatial grain bins by filtering for "Travelling" eBird observations and using the observation checklist's "distanced travelled" as a proxy for spatial grain. Four spatial grain bins were created, with observations having distances between (1) 0.15 and 0.25 km; (2) 0.45 and 0.75 km; (3) 1.35 and 2.25 km; and (4) 4.05 and 6.75 km. We spatially thinned using *spThin* [48] the records in each of the 4 spatial grain bins separately and randomly sampled 2,500 records from each group to form a final dataset of 10,000 Anna's hummingbird occurrences of variable observational grain. The temporal grains of the occurrences were coarsened to the day of observation, as in the previous case study.

We utilized 2 different environmental annotation methodologies, firstly, a record grain-specific EVI annotation using STOAT, and, secondly, a record grain-agnostic annotation using one or relatively few static EVI layers. For the grain-specific annotation, we used STOAT's Landsat 8 EVI layer, annotating with a separate spatial buffer (effective annotation grain) for each bin. The spatial buffers used were the lower bounds of the spatial grain bins (e.g., 150 m for all occurrences within the 0.15- to 0.25-km bin), with the temporal buffer set to 32 days across all observations. For the grain-agnostic annotations, we used Google Earth Engine [35] and conducted intersections between the occurrence records and spatially coarsened (from 30 m to 1 km) Landsat 8 layers. Using the original 8-day composite Landsat 8 product (GEE product: "LANDSAT/LC08/C01/T1_8DAY_EVI"), we generated (1) an aggregated long-term mean EVI layer averaging all layer data from 2013 to 2017; and (2) long-term monthly EVI averages across the years 2013 to 2017 (i.e., 1 long-term layer for each of 12 months). We conducted our grain-agnostic annotations against these 2 layers, which represent common spatial and temporal grains (1-km monthly and long-term EVI) used in ecological analyses, and which serve as a contrast to grain-specific annotation. Spatial intersections were done to the grid cell of intersection, with no spatial buffer applied. The date of observation was coarsened to the month of observation for annotation against the monthly layers or was completely ignored when annotating against the long-term EVI layer. Results from both grain-specific and grain-agnostic annotations were downloaded and analyzed in R. Data were filtered to

those with valid annotated values across all methods, resulting in 8,798 observations in total, split into 2,101, 2,175, 2,234, and 2,288 observations in the 4 spatial grain bins from finest to coarsest. Missing data were primarily attributable to cloud cover in the grain-specific annotation.

We quantified the impact of the differing annotation methods by subtracting the annotated EVI values of the grain-agnostic annotations from the same record's grain-specific EVI, generating 2 ΔEVI metrics (grain specific: monthly and grain specific: long term). We plotted the absolute value of the former ΔEVI against the original record spatial grain as we hypothesized that annotated values from the 2 methods would become less similar as record grains deviated from the common annotation grain of 1 km imposed for the grain-agnostic annotation. We plotted the second ΔEVI metric (grain specific: long term) against the Julian day of record observation to illuminate seasonal patterns in the differences between grain-specific and grain-agnostic annotation methodologies. All analyses again used R version 3.6.0 [49].

## Results

### Case studies

**Spatial and temporal scale dependence of species niches.** We used range-wide occurrence data of 2 bird species to illustrate the versatility of STOAT for environmental characterization over space and time and its utility for exploring scale-dependent relationships between organisms and their environment. We performed example environmental annotations for the migratory ruby-throated hummingbird and the tufted titmouse for a subset of available data across their full global range and the 2013 to 2017 period.

The tufted titmouse performs long-distance movements away from its northern geographic range and largely tracks the annual variation in vegetation productivity and temperature. On the contrary, the ruby-throated hummingbird is a migrant with a narrower climatic niche that is tracked throughout the year through its large geographic movements. The temporally specific annotation through STOAT (Fig 5) captures these differences for remotely sensed EVI (greenness) and LST (temperature). The environmental space occupied by the 2 species is largely overlapping in the warmer months when both are at higher latitudes. In the colder months, however, environmental associations of the 2 species diverge, as the hummingbirds migrate south and the titmice endure the winter. Further niche differences are also apparent: The hummingbirds demonstrate a closer tracking of preferred temperature range than of their preferred greenness of vegetation across seasons.

**Environmental characterization across spatial grains.** We illustrate how STOAT supports identification of the dependence of environmental niche characterizations on spatial grain size using the ruby-throated hummingbird data. Specifically, we calculated the cross-grain differences in EVI (|ΔEVI|) as the absolute difference between the EVI value of a given observation annotated at a 120-m grain and that of the same observation at a 990-m grain (i.e., 120 versus 990 m spatial buffer, both Landsat 8 EVI). Higher |ΔEVI| values would indicate greater, and lower values lesser dissimilarity between the 2 spatial buffer sizes (grains). As homogeneous environments by definition should offer similar characteristics over some distance, we anticipated that |ΔEVI| should be positively correlated with environmental heterogeneity. We found substantial differences in EVI between the 2 spatial grains (Fig 6, right-hand histogram), with |ΔEVI| exceeding 0.2 in 10.1% of records (median = 0.06). For context, mean EVI of mixed forests in the Northeastern United States (which both species partially inhabit) ranges from approximately 0.2 in the winter to 0.5 in the summer, an annual swing of only approximately 0.3 [50]. See Fig 5 for overall range of EVI in this species, typically between 0.0 and 0.9. |ΔEVI| values showed a positive correlation with the EVI-based habitat heterogeneity of the landscape (Spearman R = 0.143, $N$ = 3162).

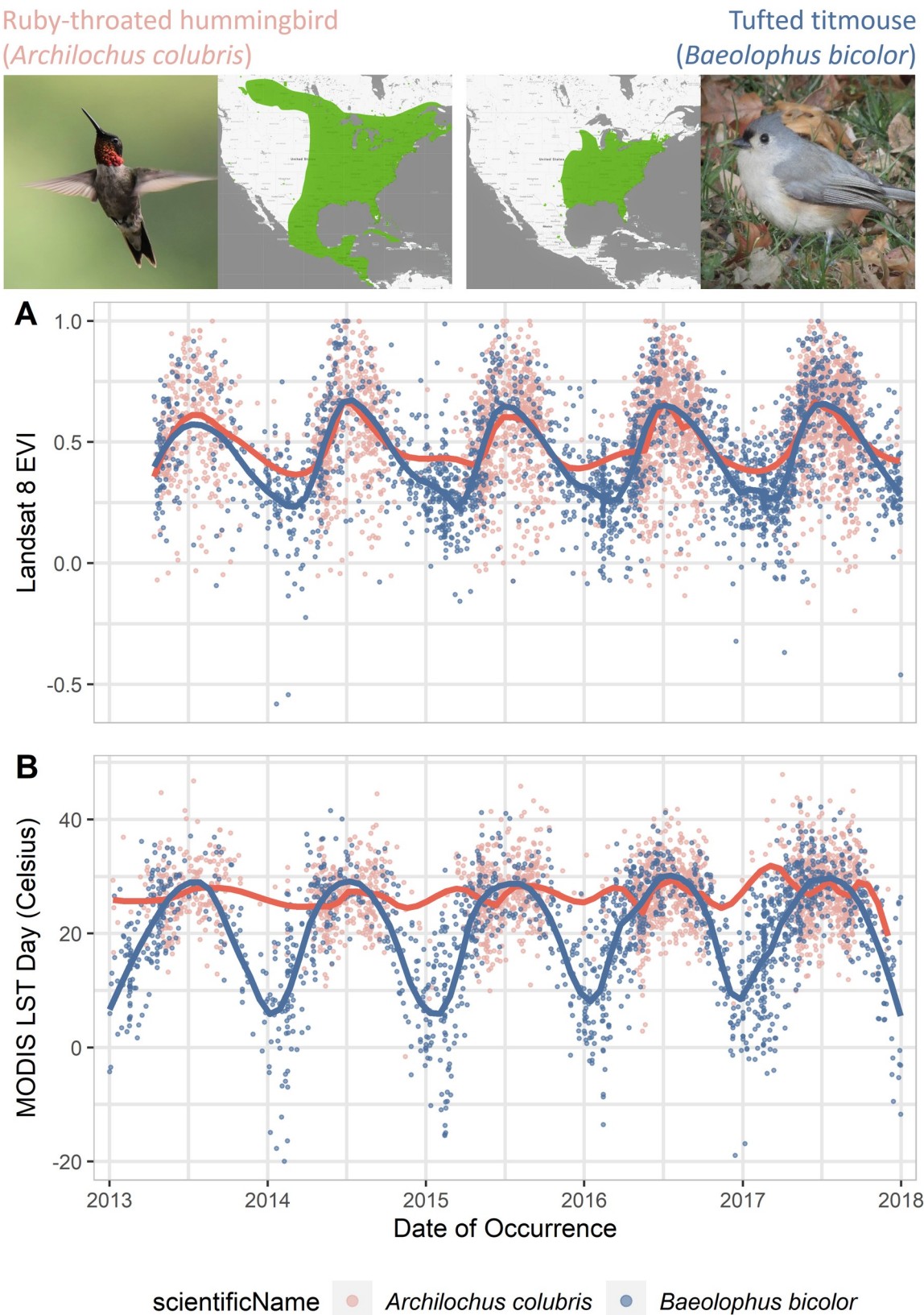

**Fig 5.** **(A)** Landsat 8 EVI and **(B)** MODIS Day LST associated with occurrences of the ruby-throated hummingbird (*Archilochus colubris*, pink) and tufted titmouse (*Baeolophus bicolor*, blue) over a 5-year period. Note the differentiation between species in occupied environmental space. A loess curve was fitted to each species for visual illustration purposes. Range maps from Map of Life (https://mol.org). The data underlying this figure may be found at https://doi.org/10.5281/zenodo.5208219. EVI, Enhanced Vegetation Index; LST, Land Surface Temperature.

**Environmental characterization across temporal grains.** We also used STOAT-based characterization of the tufted titmouse temperature niche to illustrate the effects of user-selected temporal grain (buffer). Similar to landscape heterogeneity and spatial grain above, we predicted the impact of grain to be lowest during more temporally homogeneous conditions, e.g., during times of the year with limited temperature fluctuations. We calculated $|\Delta LST|$ as the absolute value of the difference between the LST value of a given observation annotated at a 1-day grain and that of the same observation at a 30-day grain (i.e., 1 versus 30-day temporal buffer). We found substantial differences, with median $|\Delta LST|$ of 3.25 degrees Celsius and 31.5% of records exceeding a 5 degree difference (Fig 7, right-hand histogram). Confirming our prediction, $|\Delta LST|$ values were significantly larger (Mann–Whitney U test, $W = 439{,}533$, $p = 1.29\text{e-}06$) during the fall/spring "shoulder" (median = 3.59), than in winter and summer (median = 2.92).

**Grain-specific versus grain-agnostic annotation.** We compare the results of record grain-specific environmental annotations, enabled by STOAT, to record grain-agnostic annotations, which represent the status quo in most current environmental annotation applications. Specifically, we investigate the Anna's hummingbird, a species known to select areas with productive, green vegetation (high EVI) within the generally arid surrounding conditions (lower EVI) typical of its broader geographic range. Given the large heterogeneity, in space and time, of the relevant environmental conditions for this species, we expect a strong scale dependence in the environmental annotation outcome. We hypothesize annotations matching the original spatiotemporal record grain to more appropriately characterize the species' environmental requirements and predict these to be different to those using grain-agnostic standard values, e.g., strict 1-km spatial resolution and long-term annual averages.

Using a selection of records varying in original observation grain from ca. 200 m to 5,000 m, we find this expectation confirmed for spatial grain (Fig 8A). The absolute EVI difference ($|\Delta EVI|$) between matched observation grain and standard 1,000 m grain is greater than zero and increases with increasing grain gap (all using monthly layers). Specifically, the 2 grain groups closest to 1,000 m have a median $|\Delta EVI|$ of 0.038, compared to 0.051 for the farther groups (Mann–Whitney U test, $W = 11{,}158{,}301$, $p = 1.4\text{e-}35$).

To assess the effect of temporal grain mismatches, we compare results from an EVI annotation specific to the day of observation to one using a long-term annual average ($\Delta EVI$). We expect a seasonal trend in $\Delta EVI$, because the representativeness of the long-term annual average for the actually selected EVI conditions likely varies by season. This was the pattern observed, with $\Delta EVI$ peaking in the summer months (June, July, and August) compared to the winter (December, January, and February; Mann–Whitney U test, $W = 3{,}148{,}513$, $p = 2.0\text{e-}133$).

## Discussion

By offering an easy-to-use toolbox for versatile and computationally scalable environmental annotation of biodiversity data, STOAT lowers the barriers for researchers to consider the spatiotemporal grain and uncertainty aspects of their data and relevant processes. STOAT's optimized workflow and utilization of cloud-based data storage and processing allow for the efficient retrieval of environmental data. This workflow is optimized for large quantities of records significantly dispersed in space and time and custom selection of spatial and temporal

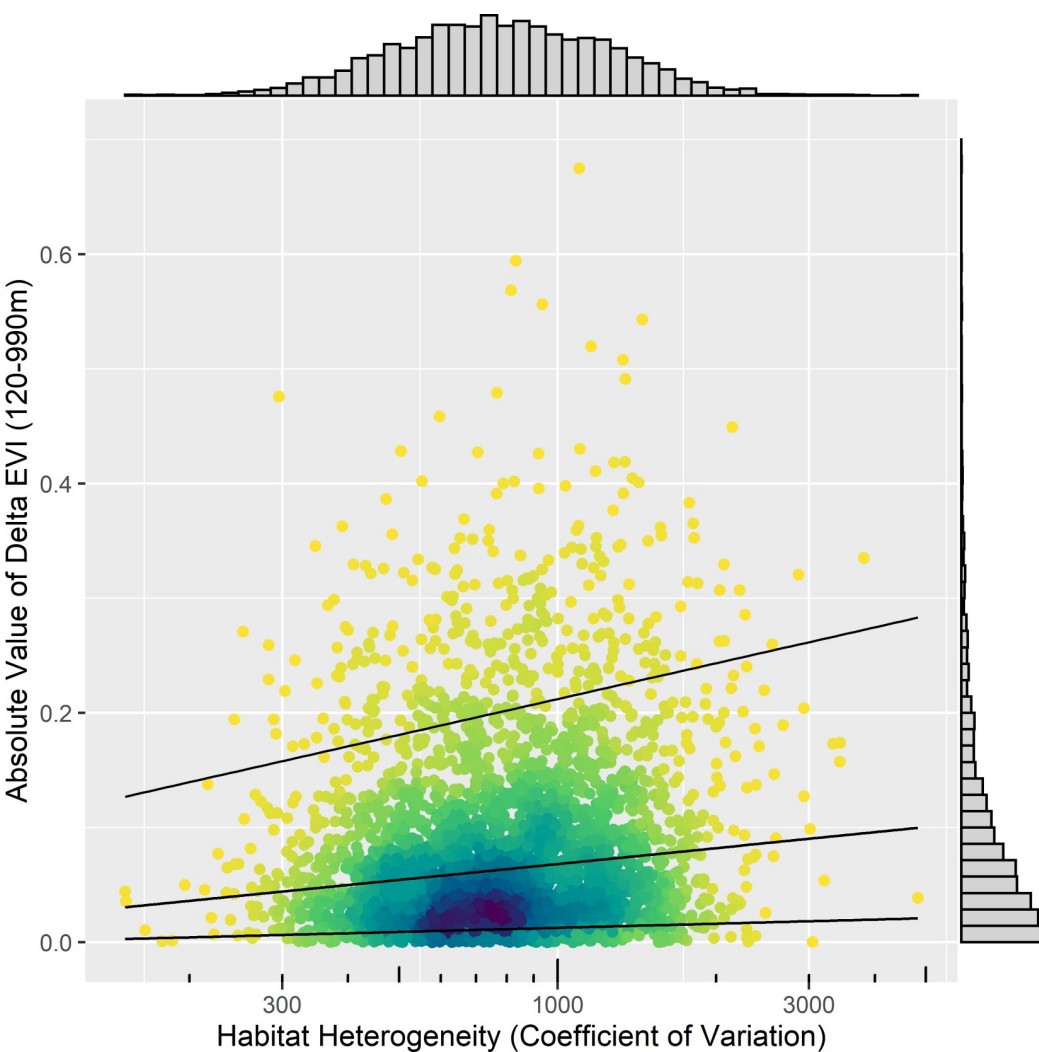

**Fig 6. Spatial grain size dependence of the ruby-throated hummingbird EVI niche across different levels of habitat heterogeneity.** The *y* axis and right-side histogram show the absolute difference in Landsat 8 EVI values (ΔEVI) between annotations conducted at 120 m and 990 m spatial grain (buffer). They are plotted in relation to a measure of landscape level (1 km) habitat heterogeneity (*x* axis, top histogram). Black lines show quantile regression fits (10%, 50%, and 90%) and illustrate how grain differences are smaller in more homogenous landscapes. Darker colors indicate greater point density. The data underlying this figure may be found at https://doi.org/10.5281/zenodo.5208219. EVI, Enhanced Vegetation Index.

buffers. STOAT reduces the complexity of environmental annotation with its user-friendly web application and R package. The larger motivation for the toolbox is to allow users to spend less time processing and linking diverse data types and more time exploring and analyzing results.

STOAT addresses key considerations of data scale and uncertainty through its spatial and temporal buffers. Buffers are a flexible tool users can apply to their specific data needs in the annotation process, including integrating data of variable grains, accounting for data uncertainty, and scale matching between data grain and extent of ecological processes. We demonstrated the scale sensitivity of species–environment relationships, both across space (Fig 6) and time (Fig 7), and illustrated the usefulness of spatiotemporal buffers in exploring these scale relationships. We explored the impacts of spatial and temporal grain on resultant EVIs (Fig 8)

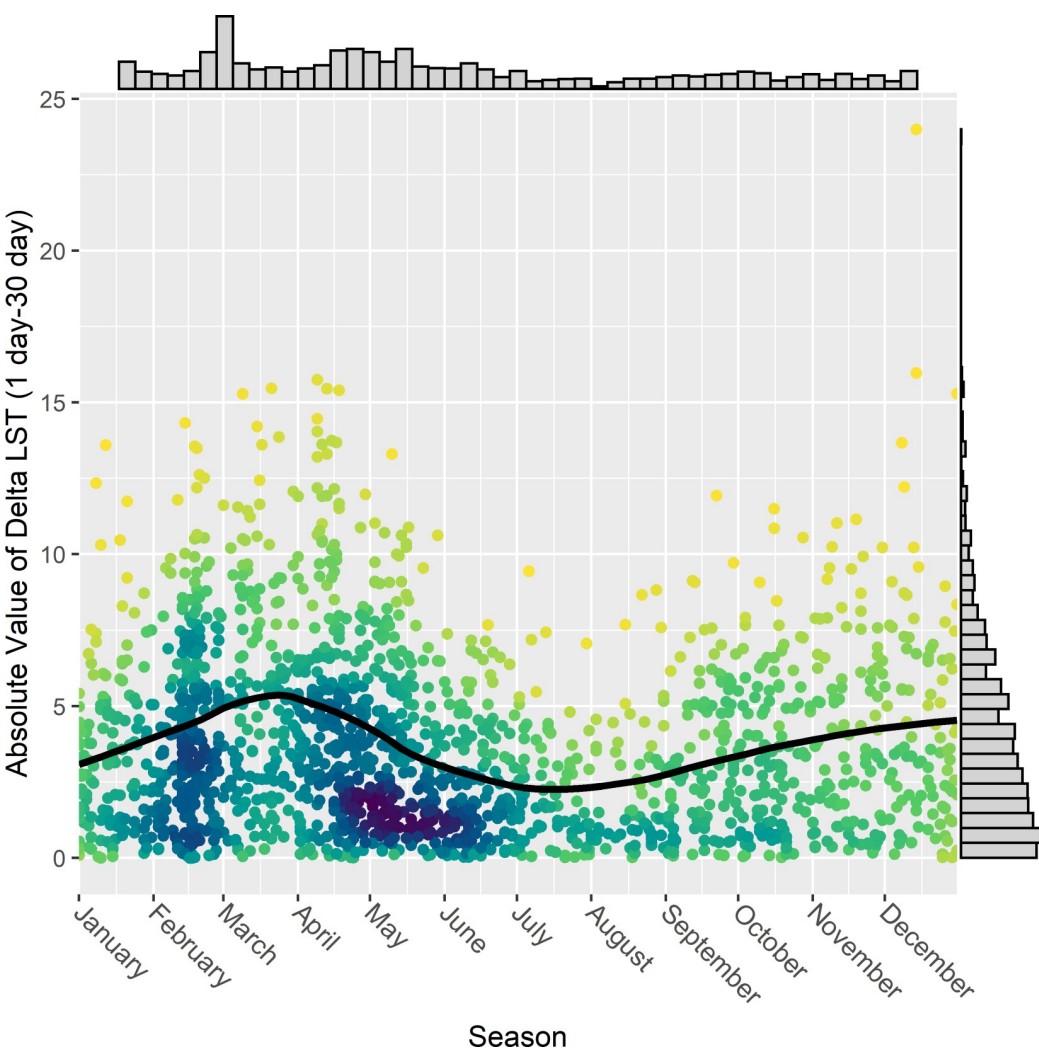

**Fig 7. Temporal grain size dependence of the tufted titmouse temperature (LST) niche across seasons.** The *y* axis and right-side histogram show the absolute difference in MODIS LST Day when annotating with a 1-day and 30-day temporal buffer. They are plotted against day of the year (*x* axis, top histogram). A fitted loess curve illustrates the seasonal trends in the temporal buffer differences. Darker colors indicate greater point density. The data underlying this figure may be found at https://doi.org/10.5281/zenodo.5208219. LST, Land Surface Temperature.

and demonstrated the need for grain-conscious environmental annotations. Scale-adjustable spatial buffering is an important tool for accurate characterization moving forward, as are finer-grain environmental datasets that allow for improved environmental characterizations.

Nonetheless, spatiotemporal buffering is not a cure-all for issues of scale. Simple buffers as currently implemented might not be able to fully characterize data needing both grain and uncertainty matching, as addressing one component of scale limits the ability to address others. One could consider data with both coarse spatial grain and high uncertainty: Using a single spatial buffer when annotating against an environmental layer, set to either grain or uncertainty, would omit the impact of the second scale component. It is not immediately clear if a simply larger buffer is appropriate to combine grain and uncertainty effects. Furthermore, computational limits necessitate upper bounds on the size of both spatial and temporal buffers. This reduces the versatility of buffers in the present, but is an unavoidable constraint given

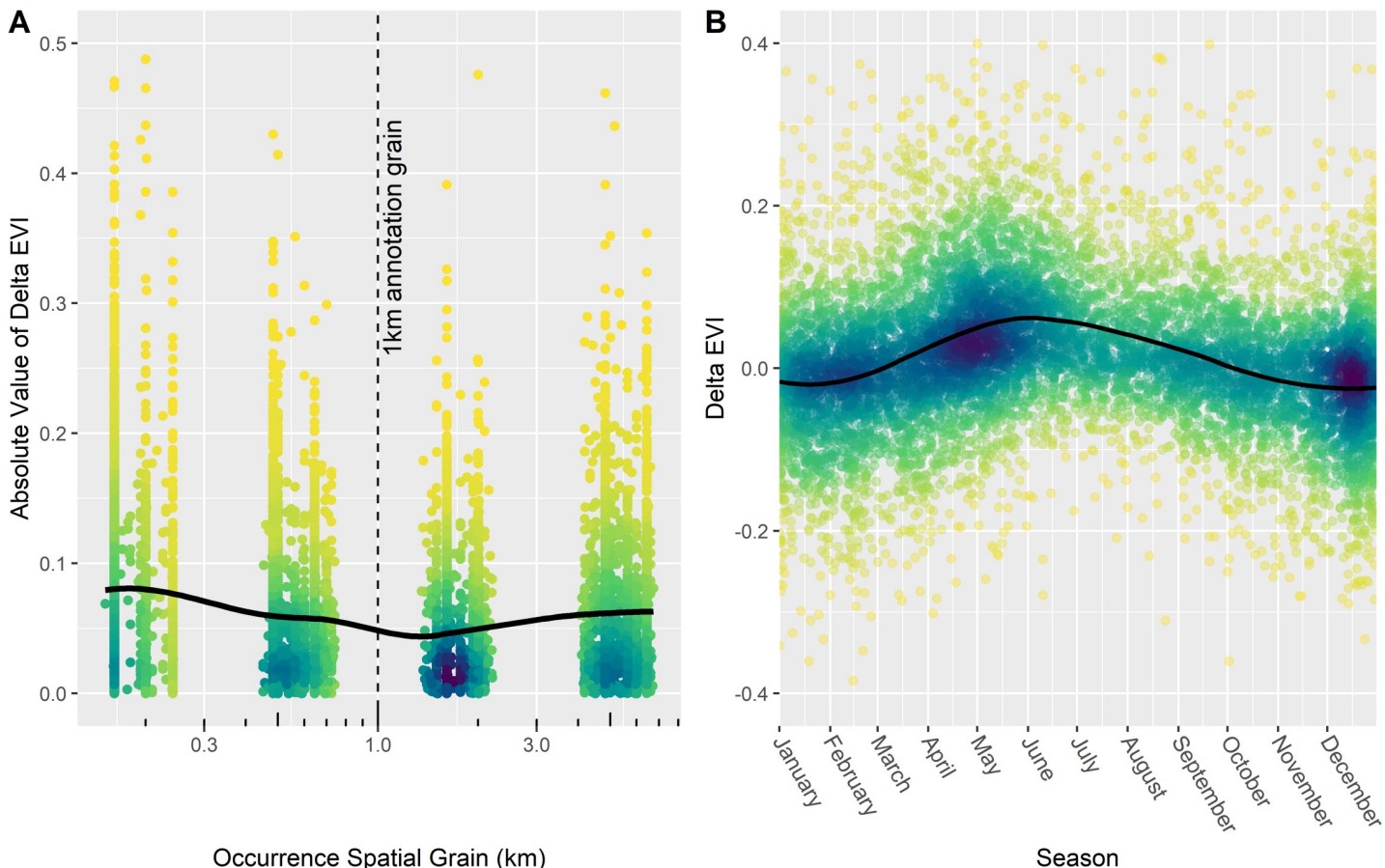

**Fig 8. Comparison of record grain-specific and grain-agnostic annotation for Anna's hummingbird.** The y axes show the difference or absolute difference in Landsat 8 EVI (*ΔEVI*) between an annotation conducted at a record's original spatial grain (and a 32-day temporal buffer) versus the same record annotated at 1 km against a monthly (A) or long-term (B) EVI layer. **(A)** |*ΔEVI*| plotted against record spatial grain. |*ΔEVI*| increases as the spatial grain of the record deviates from the standard 1-km annotation grain. **(B)** *ΔEVI* plotted against day of the year. *ΔEVI* fluctuates seasonally, reaching a peak in the summer and bottoming out in the winter. Fitted loess curves are added to illustrate both trends. Darker colors indicate greater point density. The data underlying this figure may be found at https://doi.org/10.5281/zenodo.5208219. EVI, Enhanced Vegetation Index.

finite computing resources. For spatial buffers an upper bound is particularly important, as the number of pixels in a buffered area increases exponentially with buffer radius.

## STOAT in the larger ecosystem of spatial data annotation

STOAT was developed specifically for the large-scale environmental annotation of spatiotemporal biodiversity data, providing efficient and scalable annotations beyond those previously available. Existing annotation tools have constraints that limit their ability to serve high-resolution annotations at global scale and a multidecadal temporal extent. AppEEARS [33] is optimized for the retrieval of time series data, utilizing a single temporal window across all coordinates instead of STOAT's record-specific time stamps. Users must annotate every record against the coordinate dataset's full temporal extent, and thus AppEEARS experiences severe performance reductions for datasets with broad temporal extents. Furthermore, AppEEARS does not natively support spatiotemporal buffering. Env-DATA [8] allows record-specific time stamps, being much more performant on dispersed data, but like AppEEARS provides no buffering functionality.

The extract functions available in the *raster* [32] and *rgee* [34] R packages are the most versatile existing annotators in their buffering capacity. Spatial and temporal buffers can both be implemented, although temporal buffers (spatial buffers as well for *rgee*) require user scripting. However, *raster (extract* function*)* has no product library and operates only on local layers, imposing a significant download and storage requirement on the user. This can prove prohibitive for temporally dense products like MODIS. The *rgee* (*ee_extract* function) package remedies this issue, mimicking *raster*'s functionality while simultaneously providing access to Google Earth Engine's extensive layer catalog. Nonetheless, *rgee* is constrained by a hard limit of user memory usage in Google Earth Engine, which prevents users from annotating global and temporally dispersed records against high-resolution environmental layers (e.g., MODIS) in a single extraction. Like AppEEARS, both *raster*'s extract and *rgee*'s ee_extract functions use a single temporal interval per extraction. Although the programmatic interface allows users to iteratively submit a separate extraction request for every record, this approach is prohibitively inefficient for large numbers of records. To our knowledge, currently, no tool is capable of annotating global biodiversity data at scale at the fine spatial and temporal grains that STOAT offers. STOAT is additionally currently the only annotation tool with fully integrated and computationally efficient spatial and temporal buffering, saving users' time and costs in implementation and computation.

STOAT's environmental layers span a breadth of spatial and temporal grains and extents, but may not prove sufficient for all environmental annotation purposes. On launch, STOAT's finest temporal grain layers are daily, and its coarsest is yearly. Users at scale extremes (e.g., needing hourly precision annotations) may find more suitable alternatives. Similarly, although STOAT is capable of serving precomputed layers representing long-term average environmental conditions, it is neither designed nor optimized for their ad hoc computation from fine-grain layers. Users should assess the usefulness of STOAT for their own annotation purposes by the layers available and their spatiotemporal grains. More layers will be added to STOAT over time, as we hope to make STOAT as broadly useful as possible.

## Outlook

This research and software implementation was motivated by the recognition that in the face of rapidly growing biodiversity data, researchers might benefit from a more accessible and performant way of exploring and addressing inherent issues of spatiotemporal grain and uncertainty when performing environmental annotations. In essence, all spatiotemporal biodiversity data have, and indeed vary, in their sampling grain (data resolution) and uncertainty. Alongside, the spatiotemporal scale of processes associated with environmental conditions may also vary. We demonstrated the effect of these issues on biological inference, and quantified differences between annotation methodologies, using several case studies and introduced STOAT, an easy-to-use tool that enables a more flexible environmental annotation of biodiversity data. We hope that STOAT lowers the barriers for exploring the issues of scale and uncertainty in the environmental characterization of biodiversity data and will foster a broader awareness around these issues.

## Supporting information

**S1 Table. Computational benchmarks for STOAT v1.0.**
(XLSX)

## Acknowledgments

We would like to thank David Karp for coding contributions, poet in residence Charles Marsh for suggesting the STOAT acronym, Dirk Karger for the associated provision of environmental data, and the Jetz Lab for testing and feedback.

## Author Contributions

**Conceptualization:** Richard Li, Jeremy Malczyk, Robert Guralnick, Adam Wilson, Walter Jetz.

**Methodology:** Richard Li, Ajay Ranipeta, John Wilshire, Jeremy Malczyk, Michelle Duong, Robert Guralnick, Adam Wilson, Walter Jetz.

**Project administration:** Walter Jetz.

**Software:** Richard Li, Ajay Ranipeta, John Wilshire, Walter Jetz.

**Supervision:** Michelle Duong.

**Visualization:** Richard Li, Ajay Ranipeta, John Wilshire, Walter Jetz.

**Writing – original draft:** Richard Li, Walter Jetz.

**Writing – review & editing:** Richard Li, Ajay Ranipeta, John Wilshire, Jeremy Malczyk, Michelle Duong, Robert Guralnick, Adam Wilson.

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
