## [Editor Report · Decision Letter 0]

25 Mar 2021

Dear Dr Jetz, 

Thank you for submitting your manuscript entitled "A cloud-based toolbox for the versatile environmental annotation of biodiversity data" for consideration as a Methods and Resources by PLOS Biology. Please accept my apologies for the delay incurred while we sought external advice.

Your manuscript has now been evaluated by the PLOS Biology editorial staff, and I'm writing to let you know that we would like to send your submission out for external peer review.

Please re-submit your manuscript within two working days, i.e. by Mar 29 2021 11:59PM.

Kind regards,

Roli Roberts

Senior Editor

PLOS Biology

---

## [Decision Letter · Decision Letter 1]

19 May 2021

Dear Dr Jetz,

Thank you very much for submitting your manuscript "A cloud-based toolbox for the versatile environmental annotation of biodiversity data" for consideration as a Methods and Resources paper at PLOS Biology. Your manuscript has been evaluated by the PLOS Biology editors, an Academic Editor with relevant expertise, and by two independent reviewers.

You'll see that both reviewers are broadly positive about your study, but each raises some concerns that will need to be addressed before further consideration. Reviewer#1's comments are mostly self-explanatory, but the Academic Editor asked me to reinforce the need to attend to reviewer #2's "main question", namely can you add an analysis of how much STOAT improves upon the "do nothing" approach or the current state-of-the-art? This may also help address an indirect criticism in reviewer #1's third major comment about "few quantitative contributions of this manuscript."

In light of the reviews (below), we will not be able to accept the current version of the manuscript, but we would welcome re-submission of a much-revised version that takes into account the reviewers' comments. We cannot make any decision about publication until we have seen the revised manuscript and your response to the reviewers' comments. Your revised manuscript is also likely to be sent for further evaluation by the reviewers.

We expect to receive your revised manuscript within 3 months. 

**IMPORTANT - SUBMITTING YOUR REVISION**

*Re-submission Checklist*

*Published Peer Review*

*PLOS Data Policy*

*Blot and Gel Data Policy*

Sincerely,

Roli Roberts

Roland Roberts

Senior Editor

PLOS Biology

rroberts@plos.org

REVIEWERS' COMMENTS:

Reviewer #1:

[identifies himself as Christopher Anderson]

In this work, Li et al. describe a new cloud-based system for annotating spatially-explicit species occurrence records and range map data with multi-scale, remotely-sensed environmental data. STOAT provides a browser-based UI and an open source R package, making this work accessible to a range of user expertise, and supports the analysis of custom user datasets. This system was designed to overcome several of the issues introduced by spatial and temporal scaling dynamics that affect nearly all analyses of ecological data by helping users explore a range of spatiotemporal buffering options. It is likely to become an important tool for the ecological research and biodiversity monitoring communities, and I commend the authors and developers. At this point I believe there are a few key edits to the manuscript that the authors should make in order to clarify some important issues for readers. These relate to the terminology used throughout, the potential uses of this system, and a reconsideration of the spatial scaling analysis.

First, I think it will be critical for the authors to clearly define what "annotation" refers to. I read the manuscript all the way through the first time without every discovering what it meant. It only became clear to me after I visited the R package README and read through the Github repository. It is a clear and appropriate term once the reader understands what it means, but it is not in common usage (at least within the literature I'm familiar with). I suppose I colloquially refer to it as feature extraction, or simply as zonal statistics, but these don't quite address the more comprehensive process developed for STOAT. I believe there is an opportunity for the authors to clearly define biodiversity data annotation, to state why it is important to explore data that has been annotated with multi-scale features, and to discuss the potential insights to be gleaned that will improve our understanding of biodiversity change.

The manuscript includes a strong introduction to the challenges posed by common and beguiling problems related to pattern and scale in ecology, and STOAT addresses many of them from an interesting conceptual and technological standpoint. In that sense, what the system does is clearly described. Yet it wasn't clear to me how these annotated data would be used. I had neither a sense for what other analyses these annotations would support (e.g. are these features to be included in species distribution models?) nor how the annotated data could be exported or used (the web tool appears, at the moment, to just show maps and graphs). I can imagine a few potential uses of these data - and I'm sure the authors have imagined more - but my sense was that the paper focuses more on describing the technical challenges it overcomes and less on how overcoming those challenges addresses broader problems for the community. I recommend balancing these topics.

Finally, I recommend reconsidering or reanalyzing the section characterizing variation across spatial grains; one of the few quantitative contributions of this manuscript. The proposed analysis (evaluating shifts in vegetation growth patterns (here, EVI) across spatial scales to quantify how species-environment relationships change with buffer size) is interesting and worth performing. However, from what I can tell, the author's analysis introduced a major source of variation: comparing EVI from two separate satellite sources. I suspect this is actually driving the results as presented, and needs to be addressed in a revision. There are a few sources of variation I'd like to point out.

- It is not easy to directly compare Landsat and MODIS EVI calculations, even when using harmonized ARD products. Beyond differences in radiometric calibration, signal-to-noise ratios, wavelength centers, spectral resolution and sensor-specific EVI coefficients, a major difference is in the native spatial resolution of each sensor. Due to complex radiative transfer processes, the at-sensor radiance patterns measured by a satellite changes non-linearly as the grain size of a sensor increases. In other words, resampling a group of 30x30 m measurements to 250x250 m grid cells will not necessarily match a direct 250x250m measurement over the same area. This introduces a systematic bias that is not addressed.

- The spatial uncertainty assumed by the authors (~100 m) is greater than the grain size of the fine scale measurements (30 m), meaning it's likely that many observations annotated with nearest-neighbor data are not precisely labeled. This is probably fine to a degree (first law of geography and such), and would probably average out with enough samples, but is likely to introduce a broad distribution of residuals that make it look like there's more disagreement than there really is. This is what I see when I look at the Delta EVI histogram in Figure 6.

- Oh, and there are (likely small) differences in horizontal positional accuracy (i.e. pixel location uncertainty), as well as differences introduced by the spatial projections of the data processing streams (WRS-2 for Landsat, Sinusoidal for MODIS) that would further balloon residual distributions.

I did not feel the temporal grain analysis suffered from the same issue, however, as they analyzed a consistent data source.

I recommend running a similar analysis that isolates differences between grain sizes by analyzing data that's been resampled from fine to coarse scales from a single sensor (e.g. resampling 30 m -> 240 m Landsat EVI (as the closest round multiple to 250 m)). This would obviate the differences introduced by radiometric calibration differences / etc. and create a comparison that isolates scale effects. I would probably be willing to let the spatial uncertainty issue slide if this analysis were done well and if the issue was clearly addressed in the text as a caveat.

The authors have developed a valuable cloud-based tool that I expect will be received well by the community. After revisions, I suspect this manuscript will be as well.

Below are some minor notes that don't need to be specifically addressed, but may be useful feedback for the authors.

- The R package page and the Github repository are not linked in the paper. It seems like they should be.

- Based on exploring the web UI, it looks like STOAT is optimized for working with GBIF data. I think the authors would do well to explicitly call that out to orient the reader to which ecosystem of tools this system fits into (outside of the obvious MoL connection).

- GBIF has an API, right? It seems a future version could just hook up to that instead of having users manually filter, download the csv files, then trim the fields to prepare for upload. I found this manual process to be a bit arduous and confusing (e.g., in the field reconciliation process).

- Requiring (instead of making optional) a license for uploads seems like the wrong choice: I suspect the majority of users will come here with GBIF records, which they do not have the right to assign a license to. Seems like there should be more flexibility here.

- Assuming 100 m spatial accuracy for records seems generous, but of course will vary based on the source (cell phones vs. museum records). The authors discuss this as an issue, but how it effects the interpretation of results could be more clear.

- The discussion of the EVI differences is hard to interpret. The range of values is relatively small, and not tied to any directly interpretable biophysical pattern. I felt I could use more guidance as a reader in how to understand the meaning of a median difference of 0.06 EVI.

- The last paragraph of the introduction to the discussion (l. 436-441) could be fleshed out more - it's a bit vague.

Reviewer #2:

I enjoyed the introductory section entitled Matching the respective scales of observations, uncertainty, and process. It was an excellent review in clear language without jargon, but a bit duplicative.

This section had me expecting that STOAT would allow integration of data with different spatiotemporal attributes with the environmental data layers but this is not the its function. Perhaps that can be modeled into future products to make use of more data for species with less data. STOAT seeks to properly match the spatiotemporal attributes of the biodiversity spatial data and the environmental data layers. This is important and if not done can lead to misinterpretation as stated by the authors. Any solution that makes data annotation for fitness for use easier is welcomed and STOAT appears to work well. The discussion provides realistic expectation of the data analysis which is appreciated. There is no overselling of the procedure.

My main question is: What would the data for the two case studies look like if it had not been filtered through STOAT? There is no comparison on the two case studies to a "dirty" analysis with raw data. What results would that give? This could be a strong selling point for the paper.

---

## [Decision Letter · Decision Letter 2]

8 Oct 2021

Dear Dr Jetz,

Thank you for submitting your revised Methods and Resources paper entitled "A cloud-based toolbox for the versatile environmental annotation of biodiversity data" for publication in PLOS Biology. I've now obtained advice from one of the original reviewers and have discussed their comments with the Academic Editor. 

Based on the reviews, we will probably accept this manuscript for publication, provided you satisfactorily address the following requests.

IMPORTANT:

a) I wonder if you could choose a somewhat more informative title? Maybe "A cloud-based toolbox for the versatile annotation of large-scale biodiversity data with environmental information" or some such (if that's accurate).

b) We understand that the data and code required to recreate Figs 5AB, 6, 7, 8AB are in the Zenodo deposition https://doi.org/10.5281/zenodo.5208219 - please could you cite this location clearly in each of the respective legends (e.g. "The data underlying this Figure may be found at https://doi.org/10.5281/zenodo.5208219") - you might want to include https://mol.org/stoat too, if relevant. Make sure to include the Zenodo URL in the Data Availability Statement too.

We expect to receive your revised manuscript within two weeks. 

*Published Peer Review History*

*Early Version*

Sincerely,

Roli Roberts

Senior Editor,

rroberts@plos.org,

PLOS Biology

DATA NOT SHOWN?

REVIEWERS' COMMENTS:

Reviewer #1:

[identifies himself as Christopher B. Anderson]

Li et al. have clearly addressed my comments from the previous review, and I believe this paper will be a valuable contribution to the field. I find the new structure of the introduction much easier to read, and the analysis easier to understand. Thank you for your work, and for your consideration.

---

## [Editor Report · Decision Letter 3]

27 Oct 2021

Dear Dr Jetz,

On behalf of my colleagues and the Academic Editor, Andrew Tanentzap, I'm pleased to say that we can in principle offer to publish your Methods and Resources "A cloud-based toolbox for the versatile environmental annotation of biodiversity data" in PLOS Biology, provided you address any remaining formatting and reporting issues. These will be detailed in an email that will follow this letter and that you will usually receive within 2-3 business days, during which time no action is required from you. Please note that we will not be able to formally accept your manuscript and schedule it for publication until you have made the required changes.

PRESS: We frequently collaborate with press offices. If your institution or institutions have a press office, please notify them about your upcoming paper at this point, to enable them to help maximise its impact. If the press office is planning to promote your findings, we would be grateful if they could coordinate with biologypress@plos.org. If you have not yet opted out of the early version process, we ask that you notify us immediately of any press plans so that we may do so on your behalf.

Sincerely,

Roli Roberts

Roland G Roberts, PhD 

Senior Editor 

PLOS Biology

rroberts@plos.org